



# Strong influence of solar X-ray flares on low-frequency electromagnetic signals in middle latitudes

Alexandr Rozhnoi[1], Mariya Solovieva[1], Viktor Fedun[2], Peter Gallagher[3], Joseph McCauley[3], Mohammed Y. Boudjada[4], Sergiy Shelyag[5], and Hans U. Eichelberger[4]

[1]Schmidt Institute of Physics of the Earth of RAS, 10-1 B. Gruzinskaya Str., 123242 Moscow, Russia
[2]Plasma Dynamics Group, Department of Automatic Control and Systems Engineering, University of Sheffield, Mappin 6 Str., Sheffield, S1 3JD, United Kingdom
[3]Trinity College Dublin, the University of Dublin, Dublin 2, Ireland
[4]Space Research Institute of AAS, Schmiedlstrase 6, 8042 Graz, Austria
[5]School of Information Technology, Deakin University, Geelong, Australia

**Correspondence:** V. Fedun (v.fedun@sheffield.ac.uk)

**Abstract.** In this paper we analysed Sudden Phase Anomalies (SPAs) of VLF/LF signals recorded at Graz (Austria), Birr (Ireland) and Moscow (Russia) stations during two strong solar flares in September 2017. The first X-class 9.3 flare occurred on 6 September at 12:02 UT and the second X-class 8.2 flare was observed on 10 September 2017 at 16:06 UT. Data from seven transmitters in a frequency range between 20-45 kHz are used for the analysis. The SPAs were observed in all middle-5 latitudes paths (differently orientated) with path lengths from 350 km to 7000 km. Solar X-ray burst data were taken from GOES satellite observations in the wavelength band of 0.05-0.4 nm. If was found that (i) the amplitude of SPAs in different paths varies from 10 to 282 degrees, and (ii) the correlation between the amplitudes of SPAs, the lengths of paths and the signal frequency is weak. The change in effective height of reflection due to lowering of the reflecting layer during the flares was found to be about 12 km for the first event and about 9 km for the second event. Spectral analysis of the X-ray and LF data, 10 filtered in the range between 5 sec and 16 min, showed that the LF signal spectra are very similar to X-ray spectra. Maxima of both X-ray and LF spectra are in 2-16 min interval.

## 1 Introduction

Solar flares are one of the most influential space weather events. During the flares, electromagnetic and corpuscular radiation 15 of significant power is emitted. Arrival of the solar proton radiation is mainly observed in the regions with magnetic latitudes greater than $60°$. The protons produce polar cap absorption events, which can up to last several days. In the middle latitudes, significant changes in the lower ionosphere electron density are induced by solar X-ray flares.

During most of the solar X-ray flares the electron density is strongly enhanced due to sudden increases in extreme ultraviolet and hard X-ray radiation. Ionisation is usually produced below the normal D-region (e.g., Chilton et al., 1963; Whitten and



**Table 1.** SPAs in VLF/LF signals observed on September 6, 2017.

| Path | Transmitter power, W | Frequency, kHz | Length, km | Orientation | Amax, degree | Duration, min |
|------|---------------------|----------------|------------|-------------|--------------|---------------|
| NAA-GRZ | 1000 | 24.0 | 6000 | W-E | 196 | 26 |
| NAA-BIR | 1000 | 24.0 | 4300 | W-E | 137 | 25 |
| NAA-MOS | 1000 | 24.0 | 7000 | W-E | 175 | 27 |
| NRK-GRZ | 100 | 37.5 | 3000 | N-S | 62 | 25 |
| NRK-BIR | 100 | 37.5 | 1500 | N-S | 128 | 28 |
| GBZ-GRZ | 30 | 19.58 | 1500 | N-S | 26 | 22 |
| GBZ-BIR | 30 | 19.58 | 350 | E-W | 65 | 23 |
| GBZ-MOS | 30 | 19.58 | 2500 | W-E | 80 | 27 |
| ICV-GRZ | 20 | 20.27 | 850 | S-N | 70 | 27 |
| ITS-GRZ | - | 45.9 | 1100 | S-N | 157 | 26 |
| ITS-MOS | - | 45.9 | 2700 | W-E | 177 | 26 |
| ITS-BIR | - | 45.9 | 2500 | S-N | 165 | 24 |
| TBB-GRZ | - | 26.7 | 1500 | S-N | 106 | 27 |
| GQD-BIR | 60 | 22.1 | 350 | E-W | 54 | 27 |
| GQD-MOS | 60 | 22.1 | 2500 | W-E | 75 | 27 |

Poppoff, 1965), and this region responds dramatically to the X-ray flares. Although the main source of ionisation, Lyman-$\alpha$ emission, is enhanced during the flare event, the X-ray emission overwhelms its effect several times, leading to 1-2 orders of magnitude increase in the D-region electron density (see e.g. Grubor et al., 2008).

Very Low and Low Frequency (VLF/LF) radio wave measurements are used to study the D region behaviour. These waves
are reflected by the D region, and changes in ionisation of the D-region result in amplitude and phase variations of the received VLF signal. Since Kreplin et al. (1962) have reported the relationship between X-ray bursts during flares and Sudden Phase Anomalies (SPAs), numerous works have been published on the studies of flare characteristics and SPA phenomenology (e.g., Mitra, 1974; Muraoka et al., 1977; Kamada, 1985; Khan et al., 2005). Many authors studied the ionospheric response to solar flares in the last decades (see e.g. Žigman et al., 2007; Druzhin et al., 2014; Hayes et al., 2017). Works of Raulin et al. (2010,
2013) used data from the South American VLF Network (SAVNET) to study a large number of events in order to determine the X-rays threshold, which produces ionospheric disturbances.

The boundary between the ionospheric D region and the atmosphere is characterised by two parameters: sharpness ($\beta$ in 1/km) and reflection height ($H'$ in km). For unperturbed daytime ionosphere, $\beta = 0.3$ and $H' = 71$ km (Wait and Spices, 1964). During an X-ray flare, the sunlit ionosphere reflection height decreases considerably, while the sharpness increases in
comparison with the unperturbed ionosphere. Thomson et al. (2005) modelled the D region electron densities as functions of X-ray flux up to the level of the largest, X45 flare. During this largest flare, they found that the reflecting height was about 17 km below the normal midday level. Grubor et al. (2008) investigated changes in the ionosphere during X-ray flares (ranging

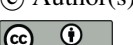


**Table 2.** SPAs in VLF/LF signals observed on September 10, 2017.

| Path | Transmitter power, W | Frequency, kHz | Length, km | Orientation | Amax, degree | Duration, min |
|------|------|------|------|------|------|------|
| NAA-GRZ | 1000 | 24.0 | 6000 | W-E | 149 | 52 |
| NAA-BIR | 1000 | 24.0 | 4300 | W-E | 122 | 45 |
| NAA-MOS | 1000 | 24.0 | 7000 | W-E | 120 | 50 |
| NRK-GRZ | 100 | 37.5 | 3000 | N-S | 282 | 63 |
| NRK-BIR | 100 | 37.5 | 1500 | N-S | 76 | 64 |
| GBZ-GRZ | 30 | 19.58 | 1500 | N-S | 20 | 50 |
| GBZ-BIR | 30 | 19.58 | 350 | E-W | 55 | 66 |
| GBZ-MOS | 30 | 19.58 | 2500 | W-E | 27 | 60 |
| ICV-GRZ | 20 | 20.27 | 850 | S-N | 47 | 50 |
| ITS-GRZ | - | 45.9 | 1100 | S-N | 98 | 54 |
| ITS-MOS | - | 45.9 | 2700 | W-E | 100 | 45 |
| TBB-GRZ | - | 26.7 | 1500 | S-N | 70 | 52 |
| GQD-BIR | 60 | 22.1 | 350 | E-W | 50 | 45 |
| GQD-MOS | 60 | 22.1 | 2500 | W-E | 52 | 47 |
| DHO-GRZ | 800 | 23.4 | 900 | N-S | 43 | 38 |
| DHO-BIR | 800 | 23.4 | 1000 | E-W | 10 | 24 |
| DHO-MOS | 800 | 23.4 | 2000 | W-E | 50 | 12 |

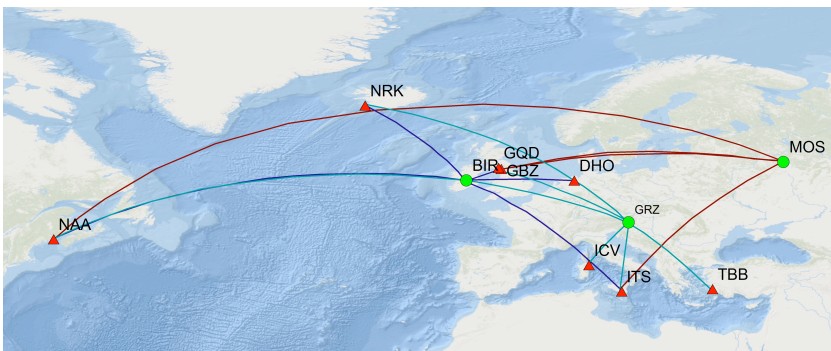

**Figure 1.** Positions of the receivers (green circles) in Birr, Ireland (BIR), Graz, Austria (GRZ) and Moscow, Russia (MOS) together with positions of the eight transmitters (red triangles), which have been used in this study.

from class C to M), which have the corresponding VLF recordings from Belgrade in 2004-2007. They found the decrease in ionosphere reflection height of up to 7 km and $\beta = 0.47$ during the strongest events.

In this work we investigate SPA characteristics from the VLF/LF signals recorded during two strong solar flares in September 2017 in the three European middle latitude VLF stations - Graz (Austria), Birr (Ireland) and Moscow (Russia). Although the





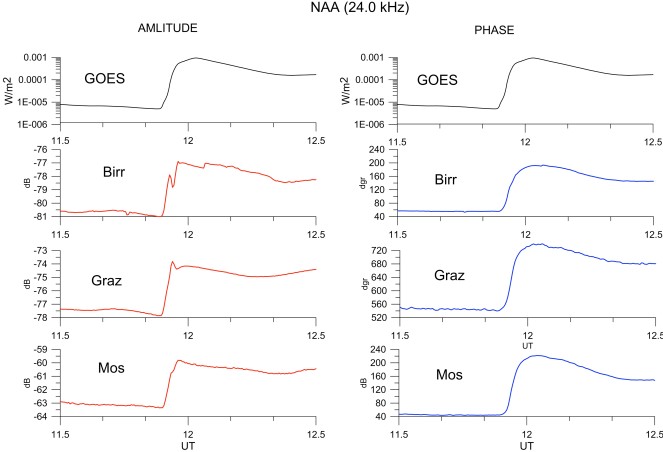

**Figure 2.** Amplitude (left) and phase (right) anomalies recorded in the NAA signal in Birr, Graz and Moscow receiving stations during the X-ray burst on September 6, 2017. Top panel shows X-ray flux measured with GOES satellite in the wavelength range of 0.05-0.4 nm.

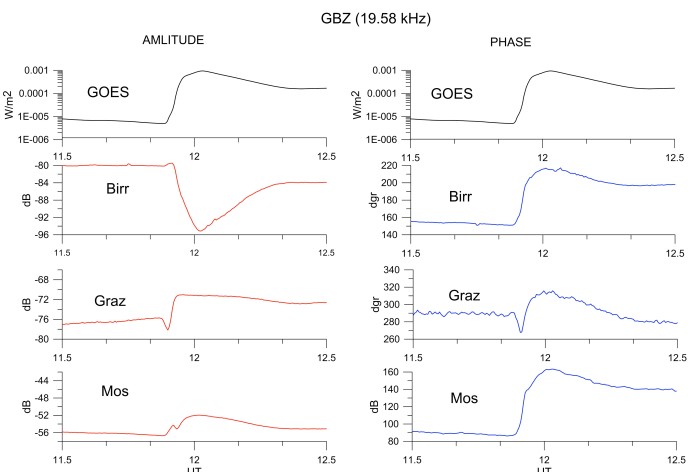

**Figure 3.** Same as Figure 2, but for the signal from GBZ transmitting station.

Sudden Ionospheric Disturbances from solar flares on the VLF signal are always recognisable well, they can vary substantially along the different way paths. The series of powerful solar flares started on September 4th from geoeffective Active Region 2673. The biggest flare to erupt from this active sunspot occurred on the 6th of September. According to the data from the geostationary satellite GOES 15 (https://www.swpc.noaa.gov/), the strongest solar flare started at 11:53 on September 6, 2017, peaked as X9.3 at 12:02 UT and ended at 12:10 UT. This was the second X-class solar flare of that day. It came just a few hours after a long-duration X2.2 at 09:33 UT.

The second strong solar flare from the same Active Region was recorded on 10 September. The event began at 15:35 achieving the maximum at 16:06 as X8.2 and ended at 16:31 UT.





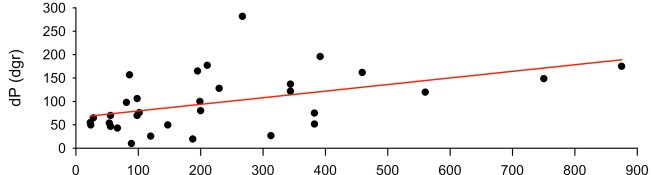

**Figure 4.** The plot of SPAs against $d/\lambda$ ($d$ is distance in km, $\lambda$ is wavelength in km).

## 2 Results of data analysis

Data from the eight ground-based transmitters in the frequency range from about 20 to 45 kHz were used for the analysis (Figure 1). The SPAs were observed in the middle-latitudes paths with lengths from 350 to 6000 km and different orientations. A typical SPA is characterised by a sudden advance of phase followed by a rounded maximum and, generally, an exponential

decay or a recovery phase. Sometimes SPAs have gradual onset. Infrequently, events have either a complex onset or complex phase variation. The classification of events as simple (S), gradual (G) and complex (C) types of SPAs was introduced by Kamada (1985). Two examples (together with amplitude variations) are shown in Figures 2 and 3.

The study of ionospheric response to solar flares remains important research, considering the possibility of improvements in the capacity and reliability of predicting disturbances in space weather, which can influence quality of radiocommunica-

tions. The strongest effect on radio wave propagation comes from the daytime low ionospheric variations. They can produce disruptions in radio communication and cause problems with navigation.

SPA characteristics for 6 September 2017 and for 10 September 2017 are shown in Tables 1 and 2, respectively. In total, 15 wave paths were analysed for 6 September, and 17 wave paths were analysed for 10 September events. The amplitude and phase anomalies due to the solar flares were observed in all the paths. The majority of anomalies on 6 September had a simple

form (exception is the path GBZ-GRZ, which had a complex onset). All of the anomalies were positive, and the shapes of SPA dependencies on time closely resemble those for X-ray irradiation. The onset time (the time difference between the beginning of the flare and the start of SPA) was about four minutes for most of the events. This is in a good agreement with (Deshpande et al., 1972) who found a delay of 3-4 minutes between the beginnings of X-ray and SPA events. The SPAs duration was on average about 26 min. The period of recovery was about 1 hour.

The SPAs on 10 September were more gradual due to the longer flare growth. All the SPAs on 10 September were positive and their duration was between 30 and 60 min. The recovery period for this day is difficult to estimate because the flare occured close to the evening terminator.

Majority of the amplitude anomalies of VLF/LF signals were positive except for 6 paths (an example of negative anomaly is shown in Figure 3).

The amplitudes of SPAs in different paths vary from 10 to 282 degrees. A weak correlation between the amplitudes of SPAs and the ratio of the path lengths to the signal wavelength was found (see Figure 4). The change in the effective height of reflection ($\Delta h$) due to the lowering of the reflecting layer during a flare has been calculated using the formulas of Westfall





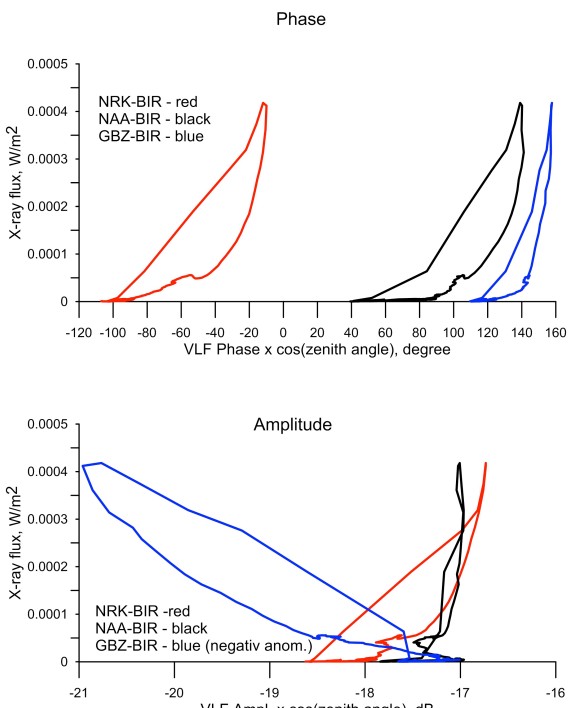

**Figure 5.** X-ray flux (irradiance) vs VLF phase and amplitude scaled with the zenith angle of the receiving station in Birr.

(1961) for the paths longer than 1000 km. It was found that $\Delta h$ varies considerably from about 2 km (GBZ-GRZ) to 20-22 km (e.g. NAA-GRZ, NRK-GRZ). On average, the lowering of the reflecting layer was about 12 km for the first event and about 9 km for the second event. In 75% of the case the decrease in the height of the reflecting layer was more than 7 km.

To establish, based on the data, a statistical relation between the additional X-ray input onto the lower ionosphere and the variations seen in the VLF paths we calculated the following parameters: X-ray flux vs VLF phase, and X-ray flux vs VLF amplitude. Both VLF phase and amplitude were scaled with the zenith angle of the receiving station. Examples for 6 September at the Birr station are shown in Figure 5.

All graphs show good correlation between the X-ray flux and VLF signals, however, the relationship for the phase is more stable. The amplitude for the path GBZ-BIR has inclination to the left because the amplitude anomaly was negative (see Figure 3). The NAA amplitude anomaly was not smooth (see Figure 2) and therefore its graph is distorted.

## 2.1 Spectral analysis

The data from GOES satellite and the phase data from the wave path NRK-BIR were used for the spectral analysis. The X-ray GOES data with the wavelength of 0.05-0.4 nm and the temporal resolution of 2 sec was filtered in the range of 5 sec-16 min. The NRK-BIR path phase data has the cadence of 1 second. The acquired spectra are shown in Figure 6. For comparison, the spectra of four undisturbed days are provided in the right panel of the figure. The spectra of the LF signals are very similar



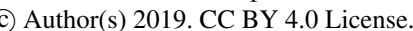

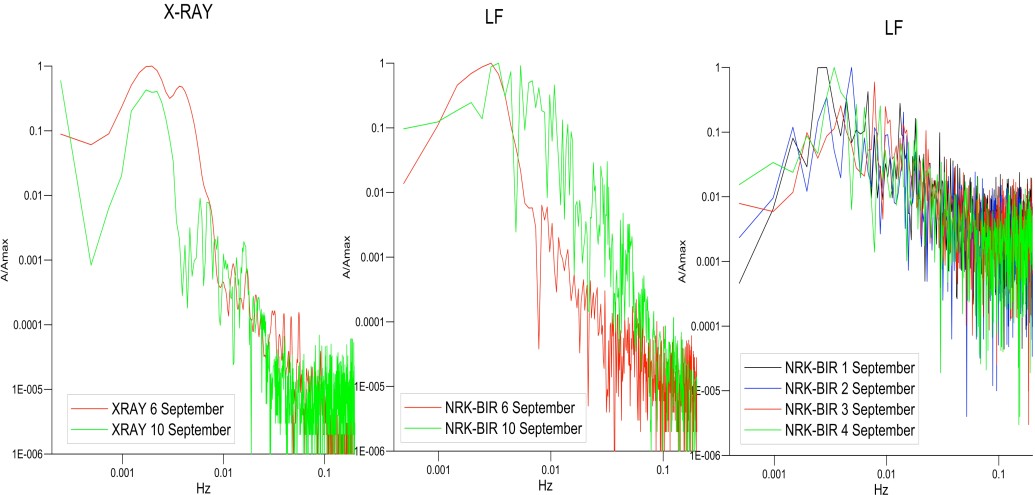

**Figure 6.** Spectra of X-ray and LF data (filtered in range 5 sec-16 min) for the two solar flares. The spectra of the same signal for quiet days are shown in the right panel for comparison.

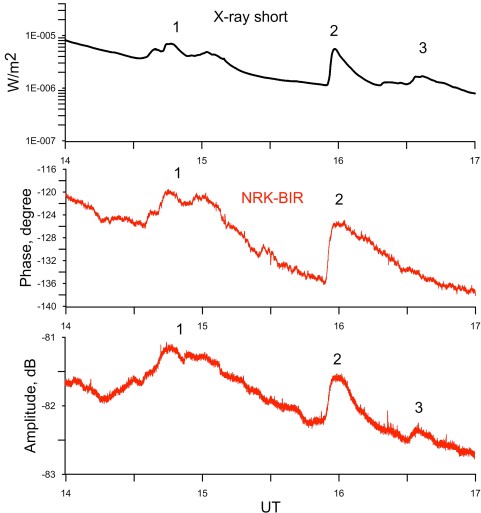

**Figure 7.** Fluctuations of X-ray flux for X9.3 solar flare on September 6 during the recovery phase (top panel) and phase and amplitude variations of the signal in the NRK-BIR path (temporal resolution is 1 s).

to the spectra of X-ray measurements. Maxima for both X-ray and LF spectra are in the interval of 2-16 min. The spectra rapidly decrease with fluctuations less than 2 min. The spectra for undisturbed days have the maxima of about 3-4 min and their decrease is insignificant.





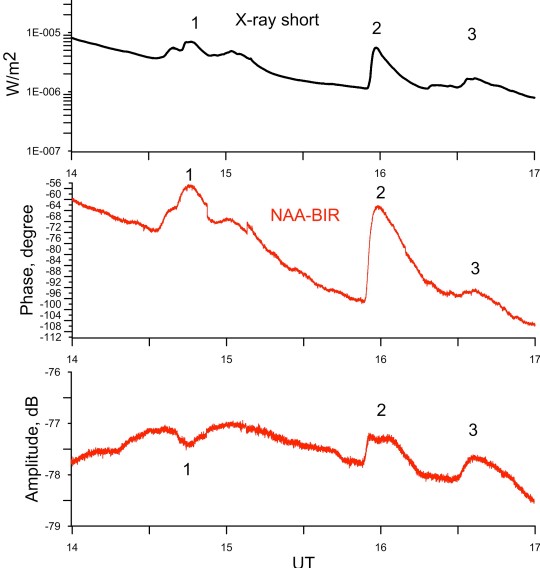

**Figure 8.** Same as in Figure 7 for the wave path NAA-BIR.

## 2.2 Response of the D region of the ionosphere to weak X-ray fluctuations during the recovery phase

Figures 7 and 8 show the influence of the weak X-ray fluctuations on the lower ionosphere. The influence can be clearly seen in the received VLF/LF amplitude and phase of the NAA and NRK signals recorded in BIR station on 6 September. The resolution of VLF/LF data in this case was 1 s.

It can be seen that VLF/LF signals repeat variations of X-ray signals. This fact is in agreement with the detection of solar flare pulsations in the Earth's ionosphere (see e.g. Hayes et al., 2017).

## 3 Conclusions

In this study, we investigated Sudden Phase Anomalies (SPAs) in VLF/LF signals (frequency range 20-45 kHz) propagating in the middle latitude paths during two solar flares with intensities X9.3 and X8.2 in September 2017. Path lengths varied from 350 km to 6000 km, and their orientation was different. In total 15 wave paths were analysed on 6 September and 17 paths were analysed on 10 September. Our main findings are as follows:

- Amplitude and phase anomalies due to the solar flares were observed in all the paths. The SPA amplitudes in different paths vary in the range from 10 to 282 degrees.

- A weak correlation between the SPA amplitudes and path lengths and the signal frequency was found.

- On average, the decrease of the reflecting layer height was about 12 km and 9 km for the first and second event respectively. In 75% of observations the decrease was more than 7 km.



- Spectral analysis revealed that the maximum of spectral density both for X-ray and LF signals was in the interval 2-16 min. The shapes of X-ray and LF signal spectra were very similar and differed markedly from those for undisturbed days.

*Acknowledgements.*   VF is grateful to The Royal Society (International Exchanges Scheme), NERC, STFC Consolidated Grant and Air Force Office of Scientific Research (USA) for support provided.



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
