# Peer review of "Strong influence of solar X-ray flares on low-frequency electromagnetic signals in middle latitudes"

_Annales Geophysicae, 2019_

## Referee Comment (RC1) · Pier Francesco Biagi (Referee) · 26 Apr 2019

The paper reports a study of the variations in the amplitude and in the phase of radio-signals lying in 20-45 kHz frequency band, during two strong solar flares occurred on September 2017. On both these occasions generally an increase of the amplitude (of different entity depending by the radio path direction), stands up; this increase is in agreement with the physical bases of the radio waves propagation. In fact, the increase is related to the fact that, on occasion of solar flares, the base of ionosphere became lower and the electron density gradient is steeper than during normal solar conditionsÂÿ thus the waves with the previous frequencies, which are reflected from

this lower layer, are more strongly reflected. In the present research, the change in effective height of reflection due to lowering of the reflecting layer during the flares was found to be about 12 km for the first event and about 9 km for the second event. In the paper, also some case of amplitude decrease is shown. I have not any explanation for this phenomenology; probably, the Authors could add some lines at this regards. In any case, the analysis reported in the paper is well conducted and the results are very interesting.

---

## Short Comment (SC1) · 29 Apr 2019

During solar X-ray flares the phases/amplitude of VLF/LF signals reflected from the ionosphere sometimes shows an opposite behavior, the so-called phase/amplitude antinomy (see e.g. Giovanni E. Perona, LF and VLF phase antinomies during solar X-ray flares, Radio Science, 10 (4), 435–444, 1975). Author has analysed two years of data and explains such behavior in terms of changes in the phase of the ionospheric reflection coefficient.

We observed antinomy (or even more complicated behavior) of phase/amplitude anomalies during solar X-ray flares on very short paths where VLF signal propaga-

tion is structured with many propagating modes, each of which has its own phase and amplitude, and the signal at any given location is the sum of those modes, unlike signal which propagates on long distances (far-field zone) where the only one mode is the principal. The distance between the GBZ transmitter and receiver at the Birr is only 350 km. It is near-field zone and in such short distance we can expect any behavior of the phase/amplitude of signal.
* * *

---

## Referee Comment (RC2) · Anonymous Referee #2 · 1 Jul 2019

Comments on "Strong influence of solar X-ray flares on low-frequency electromagnetic signals in middle latitudes" by Rozhnoi1 et al.

This paper investigate the fluctuation of VLF/LF signals response to the solar X-ray during two strong solar flares in September 2017. The observations showed that the virtual reflection height of D region, retrieved from the VLF/LF signals, was found to vary 12 km during the X-class 9.3 flare and 9 km during the X-class 8.2 flare. Further, spectral analysis of the X-ray and LF data show that the LF signal spectra are very analogous to X-ray spectra and the maxima interval between X-ray and LF spectra are in 2-16 min.

[Figure]

Generally speaking, it is an interesting paper that would enrich the recognition about lower ionosphere response to the solar flares, and the structure of the paper has been organized in a reasonable way. I have a few comments that need to be addressed, before the manuscript may be acceptable for publication.

Several major comments are suggested as follows: 1. In section abstract. Line 6 in page 1: "It" was found that. . . 2. In section 1. Line 7 in page 2: Recently, Li et al. (2019) studied the ionospheric response to solar flares based on medium frequency (MF) radar, and found that the electron density (Ne) and effective reflection height (H) had profound responses to solar flares. Liu et al. (2018) investigated the ionospheric D layer height fluctuations, during the solar flares, by analysing the VLF/LF waveforms emitted by lightning. It was found that the short-time fluctuations of D-layer height are linearly correlated to the flux density of solar radiations.

3. In section 2, page 5: What is sampling rate of your VLF receiver? Is the receiver able to continuously record waveforms? The manuscript needs to do a better job of introducing your data.

4. In section 2. Line 8-12 in page 5: "The study of ionosphere. . .with navigation". This paragraph seems to introduce the importance of investigating the ionospheric, so it is not suitable for the position in this section. It should be moved into the introduction.

5. In section 2, page 6: State the reason why the amplitude anomaly for the path GBZ-BIR is negative shown in Figure 5.

6. In section 2, page 6: Need to briefly mention how the effective reflection heights are calculated by the VLF/LF signals in manuscripts.

7. In section 2.1, page 7: "Maxima for both X-ray and LF spectra are in the interval of 2-16 min". It seems that interval between X-ray flares and LF spectra has correlation with the class of X-ray. Discuss it in this section.

8. In section 2.2, page 8: "Response of the D region of the ionosphere to weak X-ray
fluctuations during the recovery phase". What is "weak"? Be more specific.

9. In section 2.2, page 8. Need to discuss with previous work.

10. In section 1, page 1. The effective reflection height would be different retrieved by different frequency signals. The Transmitter frequency vary from 19.5 kHz to 45.9 kHz shown in Table 1. It would be useful to compare the reflection height retrieved by different frequency signals.

Li N, Lei J, Luan X, et al. Responses of the D region ionosphere to solar flares revealed by MF radar measurements[J]. Journal of Atmospheric and Solar-Terrestrial Physics, 2019, 182: 211-216.

Liu, F. F., Qin, Z. L., Zhu, B. Y., Ma, M., Chen, M., & Shen, P. (2018). Observations of ionospheric D layer fluctuations during sunrise and sunset by using time domain waveforms of lightning narrow bipolar events. Chinese Journal of Geophysics, 61 (2), 484 – 493. https:/ /doi.org/10.6038 /cjg2018K0658.

---

## Author Comment (AC1) · 5 Aug 2019

We thank the Referee for his valuable comments which have been carefully taken into account in our revision. Below we provide answers to the specific queries raised by the Referee.

Several major comments are suggested as follows: 1. In section abstract. Line 6 in page 1: "It" was found that:

Done

2. In section 1. Line 7 in page 2: Recently, Li et al. (2019) studied the ionospheric

response to solar flares based on medium frequency (MF) radar, and found that the electron density (Ne) and effective reflection height (H) had profound responses to solar flares. Liu et al. (2018) investigated the ionospheric D layer height fluctuations, during the solar flares, by analysing the VLF/LF waveforms emitted by lightning. It was found that the short-time fluctuations of D-layer height are linearly correlated to the flux density of solar radiations.

The information and references have been added to the paper.

3. In section 2, page 5: What is sampling rate of your VLF receiver? Is the receiver able to continuously record waveforms? The manuscript needs to do a better job of introducing your data.

Thank you for this remark. The information about our receivers has been added in Introduction.

4. In section 2. Line 8-12 in page 5: "The study of ionosphere: : :with navigation". This paragraph seems to introduce the importance of investigating the ionospheric, so it is not suitable for the position in this section. It should be moved into the introduction.

Done

5. In section 2, page 6: State the reason why the amplitude anomaly for the path GBZ-BIR is negative shown in Figure 5.

It is well known fact that sometimes during solar X-ray flares the phases/amplitude of VLF/LF signals reflected from the ionosphere shows an opposite behavior, the so-called phase/amplitude antinomy (see e.g. Giovanni E. Perona, LF and VLF phase antinomies during solar X-ray flares, Radio Science, 10 (4), 435–444, 1975). Author has analysed two years of data and explains such behavior in terms of changes in the phase of the ionospheric reflection coefficient. We observed antinomy (or even more complicated behavior) of phase/amplitude anomalies during solar X-ray flares on very short paths where VLF signal propagation is structured with many propagating

modes, each of which has its own phase and amplitude, and the signal at any given location is the sum of those modes, unlike signal which propagates on long distances (far-field zone) where the only one mode is the principal. The distance between the GBZ transmitter and receiver at the Birr is only 350 km. It is near-field zone and in such short distance we can expect any behavior of the phase/amplitude of signal.

6. In section 2, page 6: Need to briefly mention how the effective reflection heights are calculated by the VLF/LF signals in manuscripts.

We added the formula.

7. In section 2.1, page 7: "Maxima for both X-ray and LF spectra are in the interval of 2-16 min". It seems that interval between X-ray flares and LF spectra has correlation with the class of X-ray. Discuss it in this section.

We cannot discuss this fact because we calculated spectra only for class X of X-ray.

8. In section 2.2, page 8: "Response of the D region of the ionosphere to weak X-ray fluctuations during the recovery phase". What is "weak"? Be more specific.

Done

9. In section 2.2, page 8. Need to discuss with previous work.

Done

10. In section 1, page 1. The effective reflection height would be different retrieved by different frequency signals. The Transmitter frequency varies from 19.5 kHz to 45.9 kHz shown in Table 1. It would be useful to compare the reflection height retrieved by different frequency signals.

Yes, such analysis would be interesting. However, for the reliability of result we need to have a transmitter which transmits several signals of different frequencies and these signals should propagate along the same path. In our case we have different transmitters which transmit signals propagating in different directions along path with different

length. Therefore the comparison would be incorrect.

Li N, Lei J, Luan X, et al. Responses of the D region ionosphere to solar flares revealed by MF radar measurements[J]. Journal of Atmospheric and Solar-Terrestrial Physics, 2019, 182: 211-216. Liu, F. F., Qin, Z. L., Zhu, B. Y., Ma, M., Chen, M., & Shen, P. (2018). Observations of ionospheric D layer fluctuations during sunrise and sunset by using time domain waveforms of lightning narrow bipolar events. Chinese Journal of Geophysics, 61 (2), 484 – 493. https://doi.org/10.6038 /cjg2018K0658.

Please also note the supplement to this comment:
https://www.ann-geophys-discuss.net/angeo-2019-53/angeo-2019-53-AC1-supplement.pdf
* * *

---

## Author Comment (AC2) · 7 Aug 2019

During solar X-ray flares the phases/amplitude of VLF/LF signals reflected from the ionosphere sometimes shows an opposite behaviour, the so-called phase/amplitude antinomy (see e.g. Giovanni E. Perona, LF and VLF phase antinomies during solar X-ray flares, Radio Science, 10 (4), 435–444, 1975). Author has analysed two years of data and explains such behaviour in terms of changes in the phase of the ionospheric reflection coefficient. We observed antinomy (or even more complicated behaviour) of phase/amplitude anomalies during solar X-ray flares on very short paths where VLF signal propagation is structured with many propagating modes, each of which

has its own phase and amplitude, and the signal at any given location is the sum of those modes, unlike signal which propagates on long distances (far-field zone) where the only one mode is the principal. The distance between the GBZ transmitter and receiver at the Birr is only 350 km. It is near-field zone and in such short distance we can expect any behaviour of the phase/amplitude of signal.

Please also note the supplement to this comment:
https://www.ann-geophys-discuss.net/angeo-2019-53/angeo-2019-53-AC2-supplement.pdf

[Figure]

**Supplement:**

[revised manuscript text omitted]